# WD Repeat Domain 5 Inhibitors for Cancer Therapy: Not What You Think

**DOI:** 10.3390/jcm13010274

**Published:** 2024-01-03

**Authors:** April M. Weissmiller, Stephen W. Fesik, William P. Tansey

**Affiliations:** 1Department of Biology, Middle Tennessee State University, Murfreesboro, TN 32132, USA; april.weissmiller@mtsu.edu; 2Department of Biochemistry, Vanderbilt University School of Medicine, Nashville, TN 37232, USA; stephen.fesik@vanderbilt.edu; 3Department of Pharmacology, Vanderbilt University School of Medicine, Nashville, TN 37232, USA; 4Department of Chemistry, Vanderbilt University, Nashville, TN 37232, USA; 5Department of Cell and Developmental Biology, Vanderbilt University School of Medicine, Nashville, TN 37232, USA

**Keywords:** cancer, cancer therapy, epigenetic, PROTAC, ribosome, MYC, WDR5

## Abstract

WDR5 is a conserved nuclear protein that scaffolds the assembly of epigenetic regulatory complexes and moonlights in functions ranging from recruiting MYC oncoproteins to chromatin to facilitating the integrity of mitosis. It is also a high-value target for anti-cancer therapies, with small molecule WDR5 inhibitors and degraders undergoing extensive preclinical assessment. WDR5 inhibitors were originally conceived as epigenetic modulators, proposed to inhibit cancer cells by reversing oncogenic patterns of histone H3 lysine 4 methylation—a notion that persists to this day. This premise, however, does not withstand contemporary inspection and establishes expectations for the mechanisms and utility of WDR5 inhibitors that can likely never be met. Here, we highlight salient misconceptions regarding WDR5 inhibitors as epigenetic modulators and provide a unified model for their action as a ribosome-directed anti-cancer therapy that helps focus understanding of when and how the tumor-inhibiting properties of these agents can best be understood and exploited.

## 1. Introduction

Since the first small molecule inhibitors of WDR5 (WD Repeat Domain 5) were described a decade ago [1], the field has advanced rapidly. There has been expanding interest in the idea of targeting WDR5 for cancer therapy, a growing list of potential cancer indications, and a profound evolution of chemical matter, with orally bioavailable picomolar inhibitors now in hand for preclinical vetting [2]. Despite this progress, mainstream thinking about how WDR5 inhibitors work has, for the most part, not kept up with the pace of drug discovery. Indeed, tenuous assumptions about targeting WDR5 for cancer therapy are prevalent, and if unchallenged their impact is only likely to be amplified as these agents make their way to the clinic. Here, we evaluate the prevailing notion of WDR5 inhibitors as epigenetic modulators and present a unified model for their mechanism of action as a unique type of ribosome-directed anti-cancer therapy.

## 2. WDR5

WDR5 is a highly conserved, highly structured, and highly networked protein [3]. As described throughout this review, its participation in a multitude of tumor-relevant processes has made it a prized target for anti-cancer drug discovery. It is a core component of at least five distinct histone modifying complexes—the SET1/MLL histone methyltransferase (HMT), the NSL and ATAC histone acetyltransferases (HATs), the PRC1.6 histone ubiquitin ligase, and an embryonic cell-specific NuRD nucleosome remodeling and histone deacetylase (HDAC) complex [3]. It is a co-factor for multiple transcription factors, including Oct4, Twist1, the retinoic acid receptor, and MYC (myelocytomatosis) [3]; functions as a “reader” of modified histone H3 tails [3,4]; regulates ribosomal protein gene (RPG) transcription [5]; and associates with the anaphase promoting complex (APC) to bookmark genes for rapid reactivation after mitosis [6]. And if this was not enough, WDR5 also recruits the kinesin motor protein KIF2A to the mitotic spindle [7], promotes ciliogenesis via the stabilization of F-actin [8], and is a hub for interactions with long non-coding RNAs such as HOTTIP [9], NeST [10], ANRIL [11], and NEAT1 [12]. These activities are disparate and not all are established to the same level of rigor. But what unites most of them—apart from epigenetic roles—are two things: they were discovered in the last ten years and they receive little consideration in mainstream thinking about the mechanism of action and potential indications of WDR5 inhibitors as anti-cancer agents.

## 3. Strategies to Target WDR5

Despite many functions and dozens of binding partners, one curious aspect of WDR5 behavior is that all proteins that directly interact with WDR5 do so through one of two interfaces (Figure 1a) [3]. On one surface of WDR5 is a shallow hydrophobic cleft referred to as the “WBM” (WDR5-binding motif) site. As its name implies, the WBM site docks with proteins carrying a WBM, which includes the SET1/MLL complex member RBBP5, the NSL component KANSL2, and MYC family proteins. Partners that bind the WBM site of WDR5 do so with almost identical modalities. On the opposing surface of WDR5 is the “WIN” (WDR5-interaction) site, a deep arginine-binding cavity that engages an arginine-containing WIN motif present in proteins such as SET1/MLL family members, KANSL1, and histone H3, all of which bind WDR5 in nearly identical fashions. The recurring use of these two sites sets up mutually exclusive interactions that may help WDR5 neatly assemble into its different complexes [3], and it makes devising strategies for pharmacologically targeting WDR5 fairly straightforward: find a drug-like inhibitor of the WBM or WIN sites, or discover a PROTAC (PRoteolysis TArgeting Chimera) (Figure 1b).

A number of WBM site inhibitors have been discovered [14,19,20,21], but these have yet to progress sufficiently for thorough interrogation in cells or animal models. The discovery of WIN site inhibitors (WINi), in contrast, has been very successful (e.g., [1,2,15,16,17,22,23,24,25,26,27,28,29,30,31,32,33,34]), due in large part to the tractability of the WIN site and a deep understanding of structure–activity relationships in this class of agents. Given their prominence, advanced state, and the extent to which they have been profiled in vitro and in vivo, it is likely that WINi will dominate the field in the near future. Accordingly, this review focuses largely on this class of agents. PROTACs, or WDR5 degraders, are the most recent group of WDR5 inhibitors to emerge, all of which are built with a WIN site ligand as the WDR5-targeting entity [18,35,36,37,38]. By triggering WDR5 destruction, PROTACs should in theory supersede both WBM and WIN site inhibitors. This utility, combined with the swiftly rising tide of advancements in PROTAC technology [39], is poised to make targeted destruction of WDR5 an increasingly prominent part of the WDR5 inhibitor landscape.

## 4. The Premise and the Promise

A set of landmark papers in the early 2000s gave WDR5 its epigenetic credentials by establishing its role in scaffolding the assembly of the so-called “MLL/SET” family of HMTs [40,41,42,43]. In these multiprotein complexes, one of six MLL/SET enzymes (MLL1–MLL4, SETD1A, or SETD1B) [44] associates with a “WRAD” core (WDR5, RBBP5, ASH2L, and DPY30), as well as ancillary proteins, to catalyze histone H3 lysine four di- and tri-methylation (H3K4me2/3)—marks of transcriptionally active chromatin [45]. The enzymatic activity of MLL/SET proteins is stimulated by their association with WRAD, but in different ways depending on the family member [46,47]. MLL1 (Mixed Lineage Leukemia 1) activity as a di- and tri-methylase is dependent on insertion of the WIN motif in its carboxy-terminus into the WIN site of WDR5, and, accordingly, the catalytic activity of MLL1–WRAD complexes is profoundly and uniquely sensitive to WIN site inhibition [15,16]. In broad terms, therefore, the genesis of WDR5 inhibition for cancer therapy—reduced to practice in 2014 [1,15]—is the notion of selective inhibition of MLL1.

At that time, inhibiting MLL1 via WIN site blockade had a clear indication: MLL-rearranged leukemias (MLLr) [15]. In these cancers, one copy of *MLL1* is translocated to 1 of up to 100 different loci, leading to the expression of a fusion protein that, despite having no HMT activity, is profoundly oncogenic [44]. Widespread retention of the untranslocated *MLL1* allele in MLLr leukemias, together with studies showing that MLL-fusion oncoproteins (e.g., MLL–AF9) require a wild-type copy of *MLL1* to drive leukemogenesis [48], fueled the concept that MLL1 activity is a unique vulnerability in these malignancies. And indeed, Dou and colleagues showed that peptidomimetic WINi MM-104 is active against MLLr cancer cells in vitro and works by inhibiting H3K4me2/3 at tumor-critical MLL-fusion target genes, decreasing their expression [15]. The success of this early campaign clearly demonstrated that WDR5 is a viable target for drug discovery, energizing both the quest for drug-like WDR5 inhibitors and the search for additional venues in which they can be used to ameliorate cancer.

Since 2014, potential applications for WDR5 inhibitors have accrued rapidly. Breast cancer [49,50,51,52], C/EBPα-mutant leukemias [17], MYC-driven cancers [53], pancreatic cancer [54,55], p53 gain-of-function mutant cancers [56], colorectal cancer [57,58], neuroblastoma [5,20], hepatocellular carcinoma [59], bladder cancer [60,61], rhabdoid tumors [62], multidrug-resistant cancers [63], and glioblastomas [64] are all cancer settings in which WDR5 inhibition has been proposed as a future therapy. More recently, a wide assortment of non-cancer applications have also been proposed, including improved in vitro fertilization in cattle [65], as well as treating aliments such as chronic kidney disease [66], neuropathic allodynia [67], Alzheimer’s [68], acute kidney injury [69], cardiac fibrosis [70], and atherosclerosis [71]. Clearly, if WDR5 inhibitors can live up to this potential, their impact on human health would be profound.

## 5. A Mechanistic House of Cards

Regardless of application, most studies of WDR5 inhibitors paint them as a targeted epigenetic therapy, combatting disease by inhibiting pathogenic histone H3K4 methylation. This is the paradigm birthed by the initial work in MLLr leukemia, and it has gained considerable momentum in subsequent years. Mechanism may be trumped by safety and efficacy, but as it also sets expectations for utility and influences clinical implementation, it is worth evaluating how well the notion of WDR5 inhibitors as an epigenetic anti-cancer therapy holds up when viewed a decade from its inception.

### 5.1. Cracks in the Foundation

In 2014, there was a strong rationale for inhibiting MLL1 activity in MLLr leukemia. The field had compelling in vivo data showing that ablation of wild-type *MLL1* suppresses leukemogenesis by MLL-AF9, and there was proof of concept that MLLr cells are exquisitely sensitive to peptidomimetic WIN site blockade. But just months after MM-104 was reported, the Ernst laboratory published that the HMT activity of MLL1 is dispensable for oncogenic transformation by MLL–AF9 [72], and three years later that MLL1 itself is dispensable, with the MLL2 family member playing a major role in sustaining the disease [73]. As the HMT activity of MLL2 complexes is impervious to WIN site inhibition, including by MM-104 [15], the mechanistic premise of using WINi to treat MLLr leukemia—the premise that launched a multitude of other potential applications of these agents—cannot be correct.

### 5.2. The Importance of Family

As mentioned, genome-wide patterns of H3K4 methylation are governed by the enzymatic activity of six MLL/SET family members, all of which complex with WDR5 but not all of which depend on the availability of the WDR5 WIN site. The division of labor among the six types of MLL/SET complexes is not clearly defined, but where examined they tend to play non-overlapping roles, laying down these epigenetic marks at specific loci or in conjunction with specific co-factors [74]. Recall that the HMT activity of MLL1 complexes is uniquely sensitive to WINi [15,16]—at least under the conditions examined to date—meaning that any invocation of WINi as H3K4me2/3 antagonists has to be tied to MLL1. Unfortunately, as potential applications for WIN site inhibitors have increased over the decade, so too has the tendency to jettison this small but critical distinction. With perhaps as little as 5% of genome-wide H3K4 trimethylation catalyzed by MLL1 [75], odds are that mechanisms tying WIN site inhibitors to response-driving H3K4me2/3 changes at specific genes—without robust supporting data—will be wrong.

### 5.3. Moonlighting with WDR5

One of the most significant differences between the WDR5 ecosystem in 2014 versus that in 2024 is the burgeoning number of processes in which WDR5 has been found to function. Very few of these have been interrogated for their involvement in the response to WDR5 inhibitors. And it is likely that more processes and pathways await discovery. Our analysis of the impact of WINi on the WDR5 interactome, for example, identified more than a dozen novel proteins that are displaced from WDR5 by WINi [76], including tumor-relevant factors such as PDPK1 and mTORC2. We also found several WBM-site binding partners that load onto WDR5 when the WIN site is inhibited. Just because WDR5 is more complex than first imagined does not necessarily mean that we need to abandon the idea that changes in H3K4me2/3 underlie WDR5 inhibitor activity. But it does make it increasingly more difficult to pin the action of these agents to just a sliver of WDR5 function without a compelling reason.

### 5.4. Epigenetic Inertia

In 2019, we reported potent small molecule inhibitors that bind the WIN site of WDR5 with picomolar affinity, specifically block the HMT activity of MLL1 complexes, and selectively kill MLLr leukemia cells in culture—recapitulating much of the behavior of MM-104 [16]. Examining chromatin status, however, we saw no change in histone H3K4 tri-methylation, either globally [77] or at genes that are rapidly suppressed by WINi [16]. Clearly, these findings do not speak to whether WINi can ever inhibit cancer cells by driving down H3K4 tri-methylation, but they equally clearly demonstrate that their ability to modulate gene expression is not inexorably linked to changes in this epigenetic histone mark.

### 5.5. H3K4 Methylation—It’s Complicated

Almost all reports evoking an H3K4 methylation-based mechanism of action for WINi rely on the notion that inhibiting H3K4 di- and tri-methylation at a locus will inevitably decrease its transcriptional status. But this is a tenuous assumption. Although H3K4me2/me3 are both marks of active chromatin [45], whether they are activating marks that are instructive for transcription is much less certain. Cause and effect are difficult to separate. Many factors have stymied a clear understanding of the role of H3K4me2/3 in gene regulation—including the complexity of the enzymes involved, the interdependency of epigenetic modifications, the labyrinth of potential H3K4 methyl readers, and the failure of long-term genetic manipulations to expose a clear and consistent relationship between H3K4 methylation and transcriptional output [78,79]. Recent studies employing acute perturbations and focused assays are beginning to disentangle the relationship between H3K4 methylation and gene expression [80]. Until a consensus is reached in this area, however, drawing a straight mechanistic line between the transcriptional consequences of WIN site inhibition and changes in H3K4 methylation status is risky business.

## 6. Found in Translation: A Unified Mechanism of Action for WINi

If changes in H3K4 methylation cannot adequately explain the action of WINi in cancer cells, is there a mechanism that can? Possibly. Stepping outside the epigenetic framework, there is an alternative explanation for WINi activity that centers on the suppression of a conserved and predictable set of genes with clear ties to malignancy—genes encoding subunits of the ribosome [81].

Our studies of WIN site inhibitors in diverse cancer contexts [5,16,53,62,77,82] began to offer a unified three-phase model for the action of these agents (Figure 2). In a nutshell, protein components of the ribosome are encoded by a set of ~80 RPGs, half of which are bound by WDR5 in all cancer cell types profiled to date, including MLLr leukemia cells [16], Burkitt lymphoma cells [53], colorectal cancer cells [5], neuroblastoma cells [5], and rhabdoid tumor cells [62]. WDR5 is tethered to intronic enhancers of these genes via its WIN site [16]. One function of WDR5 at these loci is to recruit MYC, which it does by engaging a WBM found in all extant MYC family members [53]. WIN site inhibitors rapidly displace WDR5—and MYC—from these RPGs, decreasing their transcription two-fold without impacting H3K4 methylation [16,53]. Diminished transcription at target RPGs, in turn, promotes ribosome subunit attrition, choking translation and inducing a nucleolar stress response that activates p53 and its associated cell-inhibitory function [5,16,62,82]. In this scenario, the primary mechanism of action of WIN site inhibitors is the eviction of WDR5 and MYC from chromatin and select RPG suppression, and the ultimate cellular outcome is governed by whether and how cancer cells respond to what is a fairly modest decline in their ribosomal protein inventory.

Casting WIN site inhibitors as a ribosome-directed anti-cancer therapy has a number of immediate and important implications. First, it predicts that cancer cells with a high MYC burden and pristine p53 signaling will be most sensitive to single agent WINi regimens. Indeed, the involvement of MYC in MLLr leukemia, and the recurring retention of p53 in these malignancies, can explain the sensitivity of these cancer cells to WINi [83,84]. A similar scenario can also explain the sensitivity of neuroblastoma [85] and rhabdoid tumor [86] cells to these inhibitors. This mechanism also predicts additional venues in which single agent WINi activity may be expected, such as Burkitt lymphomas expressing mutant forms of MYC—which typically retain wild-type p53 [87]—as well as the large fraction of “MYC-driven” cancers (e.g., diffuse large B-cell lymphoma) that preserve p53 function on a case-by-case basis [88]. More refined patient selection criteria, including specific patterns of RPG dysregulation [89] and markers of the integrity of the nucleolar stress response [82] can also be developed. Second, it predicts that the action of WINi can be enhanced by mechanistically rationalized drug combination strategies, especially those that conspire to induce p53 or p53-dependent apoptosis in response to nucleolar stress. BET bromodomain inhibitors, for example, which suppress a distinct group of RPGs in MLLr cells, are strongly synergistic with WINi [82], as is the BCL-2 inhibitor venetoclax [82]. Other synergies are possible [62], and as our understanding of nucleolar stress signaling increases [90] more opportunities for effective drug combination approaches will undoubtedly surface. Finally, a particularly provocative prediction of a ribosome-directed model for WINi is that the “p53 barrier” to response may not be absolute—a barrier that, if broken, could dramatically expand the utility of these agents. Nucleolar stress is best understood in terms of signaling to p53, but p53-independent nucleolar stress responses do occur [91,92,93,94]. If these responses can be exposed and therapeutically exploited, the reach of WINi could readily extend to solid tumors and other malignant settings bereft of p53.

## 7. Cancer Runs on Ribosomes

Although the notion of WINi as a ribosome-directed therapy lacks the allure of epigenetic modulation, it is more firmly rooted in cancer etiology than H3K4 methylation, as connections between increased protein synthesis and malignancy are extensive [95]. The relentless rounds of cell division that cancer cells undergo must be supported by the duplication of both the genome and the proteome. MYC itself provides a frank demonstration of the importance of increased translation to cancer mechanisms. It drives the transcription of all three nuclear RNA polymerases—increasing the levels of ribosomal RNAs (rRNAs), ribosomal proteins and ribosome assembly factors, and transfer RNAs (tRNAs)—and reprograms metabolism to meet the high energy demands of ribosome biogenesis [96]. Haploinsufficiency for a single ribosomal protein is sufficient to suppress MYC-driven tumorigenesis in mice [97], and the WBM mutation in MYC, which mimics the impact of WINi on chromatin binding and RPG activation by MYC, renders MYC less able to engraft [98] and sustain [53] tumors in vivo. Add to this the frequent overexpression of RPGs in malignancy [89], the enhanced cancer risk in patients with ribosomopathies, and the prospect of “onco-ribosomes” that support pro-tumorigenic translational programs [99], and the potential of a ribosome-directed anti-cancer therapy comes into sharp focus.

Ribosome- and translation-directed therapies are not new. They are an antibiotic mainstay, and several existing chemotherapies interfere with ribosome production as part of their mechanism of action [100]. But specifically targeting ribosome biogenesis as a way to ameliorate cancer is in its infancy. Perhaps the greatest advances in this area center on the inhibition of ribosomal RNA production, a strategy driven by the discovery of agents that interfere with ribosomal RNA (rDNA) structure or RNA polymerase I activity [91,101]. These agents have undergone extensive preclinical as well as early clinical evaluation [100], and one—CX-5461—has produced significant and durable responses in select patient populations [102], albeit via complex mechanisms [103]. Within this class of agents, WINi occupy a unique place. Because the impact of WDR5 and the MYC–WDR5 interaction on target RPG transcription is around two-fold [16,62,77,82], the self-limiting nature of WINi may provide a greater margin for safety than rRNA inhibitors that have to be dosed to split the difference between ribosome addiction in normal and cancer cells. And most obviously, because WINi suppress ribosomal protein, not rRNA, synthesis, they are likely to distinguish themselves from rRNA poisons by having overlapping but distinct indications, outcomes, and pathways to resistance.

## 8. The Rise of PROTACs

PROTACs are heterobifunctional molecules that ectopically recruit a ubiquitin-protein ligase to a target protein, thereby triggering its destruction. Although PROTACs are a recent entry into the WDR5 inhibitor arena, their potential for catalytic and comprehensive WDR5 inhibition in cancer is immensely appealing. At first blush, it makes sense that if inhibiting the WIN site of WDR5 offers therapeutic benefit in some malignant settings, then clearing the entirety of WDR5 and its “oncogenic” functions [37] from a cancer cell should amplify and extend those benefits. As PROTACs gain momentum, however, we need to remember that WDR5 depletion is not the same as WIN site inhibition, and we need to understand these differences and their implications.

Side-by-side modeling of WDR5 degradation versus WIN site inhibition has demonstrated the vastly different consequences of the two approaches [77]. Ribosomal protein genes are suppressed by acute WDR5 degradation—but no more than by WIN site blockade—and MYC is evicted from chromatin at these loci [53], but that is where the similarities end. Acute degradation of WDR5 leads to wholesale loss of H3K4 trimethylation, accompanied by transcriptional changes in thousands of genes, increased anti-sense transcription, and mRNA-level changes from more than half of all active loci. These broad effects are easily rationalized by the highly networked nature of WDR5 and the notion that the destruction of WDR5 eliminates the totality of its actions and may collaterally impact other proteins or complexes that depend on WDR5 for their stability. Such consequences need careful consideration. From a therapeutic perspective, although some activities of WDR5 are neatly classifiable as oncogenic, others may not be, or their indispensability may scupper a therapeutic window for WDR5 PROTACs; this is a sharp point of concern when destroying a pan-essential protein such as WDR5 [104]. WIN site inhibition, in contrast, disrupts only a subset of WDR5 function. Some cells can clearly survive in the presence of these agents [16]. Leaving a portion of WDR5 activities intact may inherently curb the anti-tumor activities of WINi, but it may also underlie the safety of these agents and could contribute to their efficacy, especially if interaction partners that load on to WDR5 in response to WINi contribute to cancer cell inhibition [76].

It is too early to tell if WDR5-directed PROTACs can live up to their promise, or which of the two main approaches, if any, will impact cancer treatment. But as PROTACs and WINi move forward, it behooves the field to remember that these two strategies are fundamentally different and are likely to have very different uses and face very different challenges as they approach the clinic.

## 9. Conclusions

Much of the excitement in pharmacologically inhibiting WDR5 has been tied to the extraordinary potential of epigenetic modulation as a targeted and cutting-edge approach to cancer therapy. Against the enticing premise of tempering H3K4 methylation, casting WIN site inhibitors as ribosome-directed agents that work by suppressing the expression of “housekeeping” genes seems lack-luster. Yet the preponderance of evidence supports the idea that WINi act via the suppression of RPG expression, and the consistency of this response mandates that any mechanism invoked for these agents should incorporate the consequences of a declining ribosome inventory on transcript patterns, translational output, and stress, survival, and pro-death pathways. There is good reason to believe that inhibiting ribosome production can form the backbone of safe and effective cancer treatment paradigms. And, given recent advances in drug discovery, there is good reason to be excited over the prospect that WINi are a novel, unique, and viable way to get this done.

## Figures and Tables

**Figure 1 jcm-13-00274-f001:**
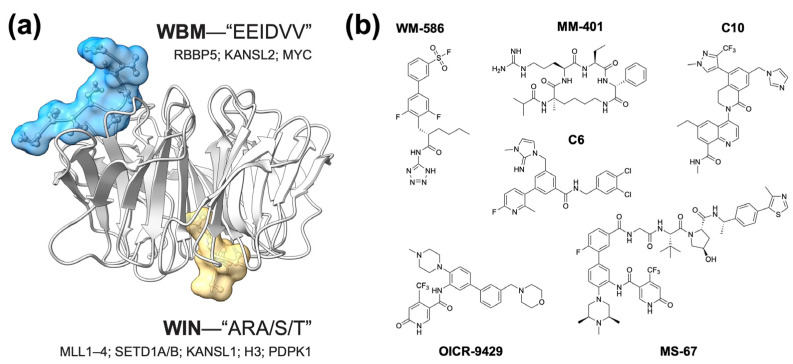
Two sites on WDR5 mediate all known direct protein–protein interactions. (**a**) Shown is the crystal structure of WDR5 (gray ribbons) in complex with a WBM-containing peptide from RBBP5 (blue surface/sticks) and a WIN-containing peptide from histone H3 (yellow surface/sticks), PDB:2XL [13]. The consensus motifs for the WBM and WIN site binding partners are shown, along with the names of representative examples of proteins that engage WDR5 via each site. (**b**) Structures of representative WDR5 inhibitors. WM-586—a covalent WBM site inhibitor [14]; MM-401—a peptidomimetic WIN site inhibitor [15]; C6—an early generation WIN site inhibitor [16]; C10—orally bioavailable WIN site inhibitor [2]; OICR-9429—one of the first small molecule WIN site inhibitors [17]; MS-67—a WIN site-directed PROTAC [18].

**Figure 2 jcm-13-00274-f002:**
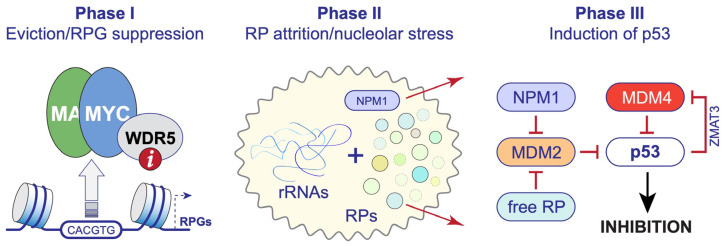
WDR5 WIN site inhibitors as a ribosome-directed anti-cancer strategy. Three-phase model for the mechanism of action of WINi. Phase I, WIN site inhibitoirs (“i”; red circle) rapidly displace WDR5 from chromatin; at RPGs where MYC binding is dependent on interaction with WDR5, MYC is also displaced. The net effect is a two-fold decrease in expression of select RPGs. Phase II, the resulting ribosomal protein (RP) attrition induces a nucleolar stress response in which free ribosomal proteins and nucleophosmin (NPM1) redistribute from the nucleolus into the nucleoplasm. Phase III, free RP and NPM1 inhibit MDM2, which results in induction of p53 and p53-dependent apoptosis. In MLLr cancer cells [82], and potentially other settings, p53 induction is amplified via activation of the splicing factor ZMAT3, which inhibits the MDM family member MDM4 (MDMX), furthering p53 function. p53-independent nucleolar stress responses may also be involved.

## Data Availability

Not applicable.

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
