# Peer review of "WD Repeat Domain 5 Inhibitors for Cancer Therapy: Not What You Think"

_jcm, 2024, doi:10.3390/jcm13010274_

Round 1
Reviewer 1 Report
Comments and Suggestions for Authors
This review, by Weissmiller, Fesik, and Tansey, considers a highly important and at the same time uncommonly complex (but meaningfully complex!) subject. The authors describe, in a very well-written paper, a set of systematic efforts, over at least a decade, to develop specific inhibitors of WDR5, a multifunctional protein that regulates transcription (among other things) and has been implicated, via several lines of high-quality evidence, as a highly promising target for anticancer therapy.
My review of the review by Weissmiller et al. can be relatively brief, since the subject matter is scientifically important while the authors’ reasoning and exposition are excellent, let alone even-handed. The emerging, significantly modified, and carefully, rigorously delineated understanding of effects of inhibiting WDR5 at its WIN protein-binding site would be of value not only to colleagues who work on WDR5, its inhibitors and related fields, but also to a much larger group of readers who are interested in challenges of modern cancer therapy.
While concentrating on the WIN binding site of WDR5, the authors also describe and discuss other ongoing fundamental and “applied” studies of WDR5, including its WBM protein-binding site and the emerging use of PROTACs to selectively target WDR5 for degradation. They point out, quite correctly (methinks), that a physical elimination of WDR5 and a selective inhibition of its WIN protein-binding site are very different things, both mechanistically and therapeutically.
I very much enjoyed reading this informative, rigorous and illuminating review of systematic and diverse studies that aim to use WDR5 as a fulcrum to selectively attack cancer cells.
The subject is fascinating. The review is excellent. My strongest recommendation.
Minor suggestions:
Line 125: “atherosclerosis” (a typo).
Line 201: “alternative” or “different” might be a better adjective, in the context, than “alternate”.
Author Response
We thank the reviewer for the positive appraisal of our review. We also appreciate them catching the typo and for the suggestion of a better adjective than "alternate" on line 201.
We have corrected the spelling of "atherosclerosis" and used "alternative" on line 201.
Reviewer 2 Report
Comments and Suggestions for Authors
In the present manuscript, the authors reviewed current strategies and technologies to target WDR5, and also pointed out the misconceptions in understanding the function of WDR5 and the MoA of its inhibitors. The authors' opinions and perspectives based on their expertise and experiences in studying WDR5 sound persuasive and informative; however, here I suggest several points that shall be considered to step on the following process of publishing.
1. The authors MUST provide what every essential abbreviation, such as gene and protein names and jargon, stands for when they are first mentioned. (i.e. WDR5, PROTAC, etc.)
2. The authors aim to provide their perspectives and opinions on targeting WDR5 as an anti-cancer therapy and address that WDR5 is a promising target for anti-cancer treatment. Therefore, they should describe the representative roles of WDR5 in association with any features of cancers, such as tumorigenesis, progression, therapeutic resistance, and relapse, hopefully with specific mechanisms (in the section "2. WDR5", from line 37). Otherwise, the readers are left unaware of the reason for and necessity of discovering the WDR5-targeted therapy.
3. The authors should italicize all the Latin words, such as in vitro and in vivo.
4. In the section "8. The Rise of PROTACs", the authors should briefly explain PROTAC (in one or two sentences) about what it is and how it acts for those unfamiliar with it.
5. The authors suggested that WINi exerts a dual impact on histone methylation and ribosomal regulation. It would be much more informative for the readers and also strengthen the idea if the authors could elaborate on the possible synergy of having such MoA and any expected safety concerns.
Comments on the Quality of English LanguageThe quality of the English language looks okay.
Author Response
We thank the reviewer for their comments and thoughtful recommendations for how the review can be improved. Our responses to each of the five points are given below.
1. We have now provided definitions for essential abbreviations. We define "essential" as those terms that are used more than once or define a concept/gene/protein/complex that is crucial for understanding of the review.
2. The reviewer has recognized an important oversight we made in the original manuscript—failing to precisely state why there is such interest in targeting WDR5 for cancer therapy. The reason for such interest in targeting WDR5 is not that it is associated with "features of cancers" in the classic sense (it is not, for example, classified as an oncogene). Rather, the interest stems from its specific molecular functions which create opportunities for therapeutic interventions in cancer. These molecular functions, and their involvement in the action of WDR5 inhibitors, are the substance of this review. In response to this comment, we have added a statement to Section 2, as requested, explicitly pointing out this reasoning and stating that it will be described throughout the review.
3. We have italicized Latin words as requested.
4. We have now explained PROTACs at the beginning of Section 8.
5. Contrary to the reviewer's statement, we do not suggest that WINi exert a dual effect on histone methylation and ribosomal regulation. To the contrary, we devote a significant part of the review to countering the argument that histone methylation is a predominant aspect of the mechanism of action of these agents. We went back and re-read the review and feel that our position on this point is clearly stated.
Reviewer 3 Report
Comments and Suggestions for Authors
Hi April M. Weissmiller,
Thanks a lot for submitting your paper to our journal, your work is amazing,but I have three questions:
1. Many abbreviation words are not explained in your work, can you explain them in your manuscript?
2. PROTAC, inhibitor of WBM and WIN are important strategies for inhibiting WDR5, do you think SiRNA and shRNA targeting WDR5 are good options for WDR5 inhibition?
3. Since HMT activity of MLL1 is dispensable for the oncogenic transformation by MLL–AF9, what do you think the possible mechanism of WINi to treat MLLr leukemia?
Author Response
We thank the reviewer for their comments. Our responses to the three points are below.
1. In response to Reviewer 2, we have now defined all essential abbreviations.
2. The reviewer raises an interesting point. To our knowledge, however, there is no significant movement afoot to clinically target WDR5 via RNAi-based mechanisms. Given that PROTACs are in development, it is likely that this status quo will remain, as there are no large conceptual advantages of shRNA- or siRNA-based over PROTACs. For these reasons, we chose not to include discussion of shRNA or siRNA in this review.
3. This is an excellent point. We had included a statement in the original version offering an explanation for the possible (non-H3K4me) mechanism of action of WINi in MLLr leukemia cells, but realize in retrospect that it was cryptically worded. In essence, viewing WINi as a ribosome-directed therapy provides an explanation for their mechanism in MLLr cells, as it predicts that cancer cells with "a high MYC burden and pristine p53 signaling" will be most responsive to WINi. MLLr cancer cells fit these criteria. We retooled this section to specifically point out the involvement of MYC in MLLr-leukemia (with references) and the recurring retention of p53 in these cancers.
Reviewer 4 Report
Comments and Suggestions for Authors
This manuscript should more enriched by figures and tables
it is too short as a review article
Moderate editing of the English language required
Author Response
The purpose of this review is to provide a pithy summary of common misconceptions over targeting WDR5 in cancer and to propose a model that can hopefully inform and influence future research in this area. We have condensed a significant body of primary and review literature (112 references) into what we hope is a concise, readable, and accessible piece that justifies our thinking in this area. We do not think it would be improved by adding bulk. Currently, excluding references, the review is ~3900 words.
Round 2
Reviewer 4 Report
Comments and Suggestions for Authors
Minor editing of English language required
Comments on the Quality of English LanguageMinor editing of English language required
Author Response
We have gone through the manuscript and edited for clarity. We have also reformatted the references to match the journal style. Finally, we have removed several of the references to our own work. The number of references to our work is now 10, out of 104 total (under 10%). Thank you